# The Role of Inflammation in Cardiovascular Disease

**DOI:** 10.3390/ijms232112906

**Published:** 2022-10-26

**Authors:** Michael Y. Henein, Sergio Vancheri, Giovanni Longo, Federico Vancheri

**Affiliations:** 1Institute of Public Health and Clinical Medicine, Umea University, 90187 Umea, Sweden; 2Institute of Environment & Health and Societies, Brunel University, Middlesex SW17 0RE, UK; 3Molecular and Clinical Sciences Research Institute, St. George’s University, London UB8 3PH, UK; 4Interventional Neuroradiology Department, Besançon University Hospital, 25000 Besançon, France; 5Cardiovascular and Interventional Department, S.Elia Hospital, 93100 Caltanissetta, Italy; 6Department of Internal Medicine, S.Elia Hospital, 93100 Caltanissetta, Italy

**Keywords:** atherosclerosis, inflammation, coronary artery disease, stroke, cerebral artery aneurysm, coronary atherosclerotic plaque

## Abstract

Atherosclerosis is a chronic inflammatory disease, in which the immune system has a prominent role in its development and progression. Inflammation-induced endothelial dysfunction results in an increased permeability to lipoproteins and their subendothelial accumulation, leukocyte recruitment, and platelets activation. Recruited monocytes differentiate into macrophages which develop pro- or anti-inflammatory properties according to their microenvironment. Atheroma progression or healing is determined by the balance between these functional phenotypes. Macrophages and smooth muscle cells secrete inflammatory cytokines including interleukins IL-1β, IL-12, and IL-6. Within the arterial wall, low-density lipoprotein cholesterol undergoes an oxidation. Additionally, triglyceride-rich lipoproteins and remnant lipoproteins exert pro-inflammatory effects. Macrophages catabolize the oxidized lipoproteins and coalesce into a lipid-rich necrotic core, encapsulated by a collagen fibrous cap, leading to the formation of fibro-atheroma. In the conditions of chronic inflammation, macrophages exert a catabolic effect on the fibrous cap, resulting in a thin-cap fibro-atheroma which makes the plaque vulnerable. However, their morphology may change over time, shifting from high-risk lesions to more stable calcified plaques. In addition to conventional cardiovascular risk factors, an exposure to acute and chronic psychological stress may increase the risk of cardiovascular disease through inflammation mediated by an increased sympathetic output which results in the release of inflammatory cytokines. Inflammation is also the link between ageing and cardiovascular disease through increased clones of leukocytes in peripheral blood. Anti-inflammatory interventions specifically blocking the cytokine pathways reduce the risk of myocardial infarction and stroke, although they increase the risk of infections.

## 1. Introduction

Over the last two decades, clinical and experimental studies have shown that atherosclerosis is a low-grade, sterile, inflammatory disease [1,2]. Systemic and local inflammation have a central role in the development and progression of cardiovascular disease (CVD), from endothelial dysfunction to clinical syndromes [3,4,5,6]. Inflammatory biomarkers have been shown to predict CVD, independently of traditional risk factors [7,8,9]. Several acute and chronic conditions, including the traditional risk factors, psychological stress, autoimmune disease, microbial and viral infections, and ageing, can activate endothelial damage and dysfunction (Table 1) [10,11,12,13,14,15,16,17,18,19,20,21,22,23,24]. In turn, this promotes a vascular low-grade inflammatory response, leading to the progression of atherosclerosis [25]. Hence, inflammation is a common mechanism linking traditional and emerging CV risk factors to the development of atherosclerosis, leading to CAD, large artery thrombotic stroke, and cerebral aneurysms [1,26,27,28,29]. All phases of atherosclerosis, from retention of atherogenic lipoproteins within the arterial wall, to plaque development and rupture, involve a complex network, including innate and adaptive immune systems, bone marrow, and spleen, which modulate the pro-inflammatory and anti-inflammatory activities of protein mediators, such as cytokines, and immune cells such as leukocytes, macrophages, and lymphocytes [30]. The role of inflammation in the atherosclerosis is confirmed by the effects of statins in reducing the CV risk. Several studies have shown that most beneficial effects of statins are due to the reduction in vascular inflammation, to some extent, independent of their lipid-lowering action [31,32,33]. Moreover, nearly one-half of patients undergoing high-intensity lipid-lowering treatment with statins in a secondary prevention trial have a residual inflammatory risk and an increased risk of major CV events, despite significant lipid-lowering effects [34,35,36,37,38]. In the last few years, the inflammatory biology of atherosclerosis has been translated into therapeutic strategies. Recent clinical trials indicated that targeting inflammation results in a lower incidence of CAD and stroke [39,40,41]. This review summarizes the current knowledge about the role of inflammation and the immune system in the development of atherosclerosis, the progression to stable and vulnerable plaque, the relationship between the central nervous system and arterial inflammatory response, the role of ageing in promoting atherosclerosis beyond a prolonged exposure to the traditional risk factors, and new therapeutic opportunities targeting inflammation to reduce the CVD burden. Although most studies refer to CAD, the relationship between inflammation and atherosclerosis in coronary and cerebral arteries is based on the same mechanisms [1,27,42].

***Endothelial dysfunction***. The vascular endothelium has a critical role in transducing the risk factors into CVD. In normal individuals, the endothelium has anti-inflammatory and antithrombotic properties, and regulates the permeability to circulating molecules and the vascular tone through the balance between the release of vasodilator substances, such as nitric oxide (NO), and endothelium-derived constrictors, such as endothelin [43]. The CV risk factors, as well as bacterial and viral infections, and environmental stress, reduce the bioavailability of NO, resulting in the loss of these protective properties, the damage of endothelial junctions, and an increase in the permeability to macromolecules. These changes lead to a subendothelial accumulation of cholesterol-containing lipoproteins which triggers a low-grade inflammatory response (Figure 1) [44,45]. Several studies have shown a strong relationship between low-density lipoprotein cholesterol (LDL-C) and atherosclerosis [46,47]. Once in the subendothelial space, LDL-C undergo an oxidation and aggregation in large complexes. Moreover, in an inflammatory environment, the lipoprotein metabolism is shifted from large- and medium-size LDL-C towards small and dense (sdLDL-C) sub-fractions with a lower affinity to the liver specific LDL-C receptor [48]. Elevated levels of sdLDL-C are associated with the increased risk of CAD [49,50]. These particles have a greater atherogenicity than larger ones because of the reduced clearance from the liver LDL receptors, resulting in a greater persistence in the blood. Hence, they are more likely to enter the arterial wall due to their small size. These properties increase the exposure of the arterial wall to sdLDL-C and favour their atherogenic intravascular modification, such as oxidation, thus making them pro-inflammatory and activating the overlying endothelium.

Despite large evidence of the causal relationship between oxidized LDL-C and atherosclerosis, many CV events which occur in individuals with LDL-C levels are currently considered to be normal, even in the absence of the conventional risk factors [51,52]. Moreover, in recent years, the large increase in the prevalence of type 2 diabetes and obesity, and the control of LDL-C with effective treatment, shifted the lipid risk profile in the population from elevated LDL-C to elevated triglyceride-rich lipoproteins (TRL) and remnant lipoproteins (RLP), which are more strongly associated with inflammation than LDL-C [50,53,54,55,56,57].

Modified lipoproteins in the subendothelial space are taken by macrophages and also by dendritic cells, which are mononuclear phagocytes “resident” in the normal arterial wall since fetal life, independently of atherosclerosis [58,59,60,61]. Activated endothelial cells and macrophages produce cytokines and adhesion molecules, such as the vascular cell adhesion molecule (VCAM)-1, intercellular adhesion molecule (ICAM)-1, and E-selectin, on the endothelial surface of the artery. Circulating monocytes originating from the bone marrow or the spleen, adhere to the endothelial layer, migrate into the intima by diapedesis, and differentiate into macrophages [2,62,63]. 

Additionally, endothelial cells can undergo an endothelial-mesenchymal transition (EMT) and migrate into the intima, thus contributing to intimal thickening and inflammation [64,65]. These changes are the first step in the development of atherosclerosis, preceding angiographic or ultrasound evidence [30,45,66,67]. 

The endothelial inflammatory response includes the coordinate activation of both innate immunity (macrophages) and adaptive immunity (T- and B-lymphocytes, dendritic cells) [44,68]. Leukocytes involvement in inflammation and atherosclerosis has also been shown by human positron emission tomography (PET) studies, using ^18^F-fluorodeoxyglucose (^18^F-FDG), a glucose analogue extensively used as a marker of metabolic activity, for the malignancy staging. It is used in vascular inflammation imaging because it accumulates mostly in macrophages due to their high glucose metabolic activity, especially after an inflammatory activation [69,70]. The increased uptake has also been found in the bone marrow and spleen of patients with CAD compared with those without. This confirms the association between bone marrow and spleen hematopoietic activation and an increase in the proinflammatory mediators involved in atherosclerotic plaque inflammation [71,72,73].

Once entering the subendothelial space, the recruited monocytes differentiate into macrophages and then polarize, adopting different functional phenotypes, in response to their microenvironment [74]. T lymphocytes activate these cells into pro-inflammatory M1 macrophages, which elaborate pro-inflammatory cytokines (interleukin IL-1α, IL-1β, IL-6, IL-12, IL-15, IL-18, and the tumour necrosis factor (TNF)-α) involved in atherosclerosis progression, or alternative anti-inflammatory M2 macrophages which elaborate anti-inflammatory cytokines (IL-4, IL-10, IL-13, and the transforming growth factor (TGF)-β), which have a critical role in the resolution of inflammation and plaque healing [75,76,77,78,79]. Some interleukins (IL-1β, IL-6, and IL-12) control the hepatic production of the C-reactive protein (CRP), the most established inflammatory biomarker of CV risk [80,81,82,83]. Although macrophages are the main source of cytokines, other cells, such as lymphocytes, endothelial cells, and polymorphonuclear leukocytes contribute to their production. 

Most components of the immune system can produce pro-inflammatory or anti-inflammatory soluble factors and cells depending on the inflammatory environment. Therefore, the atheroma progression is determined by an imbalance between the pro-inflammatory and anti-inflammatory activities of immune cells [84,85]. This accounts for the dynamic progression of atherosclerotic lesions, which occurs through phases of quiescence and flares of activity triggered by systemic or regional inflammation [1,5].

***From systemic inflammation to focal atherosclerosis***. Although atherosclerosis is associated with systemic CV risk factors and systemic inflammation, atherosclerotic plaque formation has a focal distribution, predominantly at the arterial bifurcation or side branches, which are exposed to a non-uniform, disturbed blood flow (Figure 2, Figure 3 and Figure 4) [86,87,88]. This pattern of flow generates low wall shear stress (WSS) which induces vascular inflammation and drives the atherosclerosis pathology and plaque progression [89]. WSS is the tangential force of the mechanical friction of the flowing blood which acts longitudinally on the endothelial surface of the arterial wall [90]. Specific endothelial biomechanical receptors such as glycocalyx, a proteoglycan layer which covers the apical surface of the endothelial cells, sense and distinguish the laminar and non-uniform patterns of blood flow, translating WSS into biochemical signals [91]. A uniform, laminar flow induces the secretion of NO, which regulates the arterial tone, in order to maintain the anti-inflammatory and antithrombotic properties of the endothelium. Conversely, decreased WSS induces the expression of endothelial genes, controlled by flow-responsive endothelial microRNAs (miRNA), such as miRNA 92a, 663, 712, promoting the production of adhesion proteins and other inflammatory molecules that recruit leucocytes and direct their migration into the arterial wall [86,92,93,94]. This mechanism may also explain why local inflammation episodes, remote from atherosclerotic lesions, stimulate an inflammatory activation and coronary plaque progression.

***Inflammation in coronary plaque development***. Macrophages catabolize the oxidized LDL-C within the arterial wall, forming the cholesterol-laden foam cells. Depending on the inflammatory cytokine activity and the amount of oxidized LDL-C, macrophages undergo apoptosis [77]. Dead macrophages coalesce into a lipid-rich necrotic core which stimulates the migration of vascular smooth cells into the intima, encapsulated by a collagen fibrous cap, leading to the formation of fibro-atheroma, generally a stable lesion [95]. In the conditions of chronic inflammation, macrophages exert catabolic effects that degrade and thin the fibrous cap, resulting in a thin-cap (<65 µm) fibro-atheroma (TCFA) (Figure 5 and Figure 6) [85,96]. These pathological changes, characterized by large lipid-rich necrotic core separated from the arterial lumen by a thin fibrous cap, make the plaque unstable and prone to rupture, leading to thrombosis [97,98]. In turn, thrombosis also promotes inflammation through the release of inflammatory mediators from platelets [2]. As the plaque grows, the arterial wall undergoes an outward enlargement, due to a WSS increase at the site of the luminal narrowing. Initially, such (positive) expansive remodelling allows for maintaining a normal blood flow. However, in more advanced stages, the arterial wall deformation activates a further inflammation and lipid accumulation, making the plaque more prone to rupture [99,100]. Arterial remodelling due to WSS changes is also responsible for the development of a cerebral aneurysm (Figure 7) [28,101]. Inflammatory changes within the plaque make it hypoxic, leading to the development of neovascularization originating from adventitial vasa vasorum. This process contributes to plaque vulnerability [102].

***Plaque calcification***. Inflammation also stimulates the development of calcifications within the necrotic lesion as a healing response to the macrophage’s inflammatory activation [103,104,105]. Longitudinal imaging studies, using PET, have shown that inflamed arterial sites undergo the subsequent deposition of calcium, and within the same arterial segment, different degrees of inflammation show different rates of calcium deposition [106]. The death of macrophages and smooth muscle cells release vesicles acting as nucleating sites for the deposition of hydroxyapatite crystals which can aggregate, resulting in microcalcifications less than 50 µm in diameter being embedded in the fibrous cap [107,108]. Plaque calcification further stimulates macrophage infiltration, thus increasing the nucleating sites and new calcification [109]. If inflammation persists, there will be subsequent cycles of monocytes infiltration which differentiate into macrophages, that undergo death, leading to microcalcification development [110,111]. Along with TCFA and macrophage, microcalcifications strongly contribute to plaque instability, especially when they co-localize with macrophages in the same plaque (a reciprocal distance less than 100 µm), as demonstrated by optical coherence tomography (OCT) [112,113,114,115].

In addition, to further stimulate the inflammation around the lesion, microcalcifications exert a mechanical stress within the fibrous cap [116]. Biomechanical studies have shown that a plaque rupture may occur as a consequence of large stress at the interface of tissues with a different stiffness, such as hard microcalcifications within the much softer layer of the fibrous cap (large modulus mismatch) [113,117]. In accordance with this effect, the risk of a plaque rupture is proportional to the extent of the interface area [118]. In an early stage of inflammation, microcalcifications are sparse and the risk is low. As long as inflammation persists, their number increases as well as the extent of the interface between the rigid and soft regions. Over time, some of them merge into larger, dense calcified sheets of macrocalcification which have a smaller interface area and a reduced risk of rupture, thus progressing from a high-risk lesion to a more stable plaque [116,119,120]. Additionally, macrocalcifications tend to limit the extent of inflammations, as it does in other inflammatory conditions such as tuberculosis [116,121]. These observations indicate that plaque vulnerability is inversely proportional to the extent of the calcifications and account for the paradox of an improved clinical outcome despite the highly calcified arterial plaques. The extent and composition of calcified coronary artery plaques have different clinical implications. Despite this, plaque calcification is considered to be a marker of plaque stability, a direct quantitative assessment of coronary artery calcium (CAC) with an Agatston score measured by non-contrast computed tomography (CT), has consistently shown to be a strong predictor of CV events and the total plaque burden [122,123]. However, while the CAC volume is directly associated with the subsequent CV events, the association between the CAC density and the CV events is inverse [124,125]. It can be assumed that less densely calcified plaques correspond to more inflamed, lipid-rich plaques in an early stage of development, hence they are unstable. This corresponds to the decrease in the CV events induced by the statins, which are known to reduce vascular inflammation while increasing the plaque calcification and thickness, thus promoting plaque stability [126,127,128,129,130,131,132,133]. 

***Vulnerable plaque***. Inflammation is a critical feature of vulnerable plaques, although lesser degrees of inflammation have also been found in stable ones [134]. Atherosclerotic plaques consist largely of an extracellular matrix (ECM), including collagen, elastin, proteoglycan, and glycosaminoglycan, synthesized by smooth muscle cells in the arterial wall. The ECM is interlinked with plaque calcification, both contributing to the plaque stability [135]. Microcalcification localize between the collagen fibers and the regions lacking collagen, so that the proportions of the ECM and microcalcification are inversely related. Although interstitial collagen is a plaque stabilizing factor, it also contributes to the accumulation of lipoprotein particles within the arterial wall [119,136]. In conditions of inflammation, cytokines (IL-1β, TNF-α) induce the secretion of metalloproteinases (MMPs), especially MMP-1, MMP-8, MMP-9, MMP-12, and MMP-13, from macrophages, controlled by microRNAs [137,138,139]. MMPs catalyse the breakdown of the interstitial collagen, resulting in the thinning and weakening of the fibrous cap, thus compromising its tensile strength and making the plaque unstable [140]. 

Additionally, the stability of the fibrous cap depends on collagen fibre cross-linking, which is modulated by the enzyme lysyl oxidase (LOX) expressed by the endothelial cells [141,142]. High LOX levels are associated with plaque stability and the healing process within the plaque [143]. Endothelial dysfunction induced by the CV risk factors and mediators of inflammation, such as macrophages derived cytokines, reduce the LOX activity, resulting in abnormal collagen cross-linking. This process weakens the fibrous cap and increases the soluble forms of collagen which may undergo MMP degradation (Figure 8).

Statins have been shown to inhibit the secretion of MMPs from inflammatory cells and normalize an endothelial LOX expression, thus increasing the plaque collagen [144,145,146,147,148]. Therefore, in addition to the lipid-lowering effect and the increase in the calcium content of the atherosclerotic plaque, these anti-inflammatory effects account for the plaque stabilization induced by the statins.

The concept of vulnerable or high-risk plaque derives from autopsy studies showing that the rupture of the thin fibrous cap of TCFA exposes the necrotic core to the circulation blood, triggering acute thrombosis, such as a myocardial infarction and stroke [149,150]. The rupture of the fibrous cap of a TCFA has been implicated in about two thirds of acute coronary events [96,151,152]. Hence, in recent decades, great efforts have been made to identify and treat high-risk plaque. However, treating individual plaques showing “vulnerable” characteristics with a coronary stent did not reduce the risk of a myocardial infarction [153,154,155]. Moreover, in patients with myocardial infarction and multivessel disease, a complete revascularization of the culprit and non-culprit lesions with a percutaneous coronary intervention (PCI), compared to the PCI treatment of culprit-lesion-only, substantially reduced the risk of the subsequent coronary events [156,157,158,159,160]. In contrast with the post-mortem observations, human intravascular imaging studies have shown that most TCFA do not cause clinical events [161,162,163]. Imaging studies in patients with CAD have shown plaques in different stages of development, coexisting in the same artery [164,165,166]. Moreover, their morphology may change over time, spontaneously or while on statin therapy. Most progressively shift from high-risk lesions to more stable, calcified plaques, while others undergo a subclinical rupture followed by their healing, resulting in a progressive coronary lumen obstruction [167,168]. Hence, vulnerable plaques at a high risk of triggering thrombosis cannot be distinguished from the many others which will not cause clinical events. According to the current evidence, efforts to identify and treat only vulnerable plaques may be misleading. 

In recent decades, another pathophysiological mechanism that triggers plaque disruption and thrombus formation, known as endothelial erosion, has been found to account for an increasing proportion of acute coronary syndromes (ACS) [169,170,171,172,173]. Plaques undergoing superficial (endothelial) erosion show less lipid accumulation, a smaller necrotic core, fewer inflammatory cells, and an intact fibrous cap rich in collagen [174]. The thrombi derived from a superficial erosion are white or platelet-rich, in contrast with red or fibrin and erythrocyte-rich thrombi associated with the plaque rupture. These pathophysiological changes are reflected in a shift of the clinical presentation of ACS. While patients with an ST-segment elevation myocardial infarction (STEMI) associate more commonly with a plaque rupture, those with a non-ST-segment elevation myocardial infarction (NSTEMI) show a much higher prevalence of erosion [170]. In recent years, the clinical presentation of ACS has shifted from STEMI to NSTEMI, even taking into account the introduction of more sensitive assays for troponin and the reclassification of unstable angina into NSTEMI. This trend is probably accounted for by the changes in the CV risk factors due to the widespread use of statins.

Whatever the mechanism, plaque rupture or erosion, the atherosclerotic plaque instability is not only related to intrinsic plaque vulnerability. Rather, the systemic factors which influence the coagulation system, such as a systemic or local inflammation, and recurrent infections, in addition to the conventional CV risk factors, increase the risk that plaque disruption occurs in a pro-thrombotic environment [152,175,176].

Although high-risk plaques do not necessarily identify future culprit lesions, they may be associated with extensive atherosclerotic lesions. The pathophysiological role of systemic inflammation in plaque instability accounts for the frequent finding of high-risk plaques at multiple distant arterial sites, known as *multifocal coronary plaque instability* [177,178,179,180,181]. These observations support the concept that high-risk lesions, closely associated with systemic disease and extensive atherosclerotic lesions, indicate a vulnerable patient rather than a vulnerable plaque.

***Inflammatory response to mental stress***. The brain response to environmental stimuli may increase the risk of CV disease through increased inflammation, mediated by the autonomic nervous system. Acute and chronic psychological stress are frequently experienced in everyday life as anger, fear, job-strain, depression, financial problems, and loneliness [182]. Neuroimaging studies have shown that psychological stress is associated with an increase in the metabolic activity of the central autonomic network (CAN), an anatomically and functionally interconnected brainstem and the subcortical areas, including amygdala, hypothalamus, hippocampus, and thalamus, currently referred to as the limbic system. These areas are strictly connected with cortical regions, such as the medial prefrontal cortex and insular cortex, into a cortico-limbic functional network [183]. These cortical and subcortical brain regions regulate the stress perception and emotional response through a sympathetic and parasympathetic autonomic nervous system [184,185]. Neuroimaging studies have also shown that the increased metabolic activity of these areas, especially amygdala, predicts the development of CAD independently of the traditional CV risk factors [186]. In normal conditions, the vascular system is under tonic inhibitory control by the parasympathetic system in dynamic balance with the sympathetic system. The parasympathetic (vagal) efferent innervations release acetylcholine which inhibits the release of inflammatory cytokines, including the tumour necrosis factor alpha (TNF-α) and interleukins (IL-1, IL-2, IL-6), by tissue macrophages. This cholinergic anti-inflammatory pathway, known as “*parasympathetic inflammatory reflex*”, modulates the inflammatory response [187,188,189,190]. An exposure to emotional stress results in an autonomic imbalance with an increased sympathetic output and the withdrawal of the parasympathetic tone, thus leading to the release of inflammatory cytokines [191,192]. These directly impair the endothelial function, inhibiting nitric oxide (NO) synthesis and increasing the endothelin-1 (ET-1) release [45]. 

In addition to the response to environmental stimuli, the brain is thought to manage information from atherosclerotic lesions, hence modulating their progression. The adventitia of atherosclerotic arteries is innervated by sensory and sympathetic fibres, along with the aggregates of immune cells, known as *neuroimmune cardiovascular interfaces* [193]. The density of neural fibres correlates with the plaque size. This artery-brain circuit suggests that the plaque-induced activation of sensory neuronal fibres on arterial adventitia leads to the activation of hypothalamic nuclei which are involved in the sympathetic outflow, resulting in the neural regulation of plaque progression. This is confirmed by the experimental observation that the disruption of sympathetic fibres reduces the density of adventitial sympathetic nerve fibres, the aggregates of immune cells, and the plaque volume [194].

***Ageing, bone marrow activation, and clonal haematopoiesis***. Ageing is associated with an increased risk of CV disease. In addition to the burden of a long-term exposure to the CV risk factors, there is a direct relationship between ageing and low-grade systemic inflammation, leading to atherosclerosis. Bone marrow hematopoietic stem cells (HSCs), which reside in a specialized microenvironment known as the HSC niche, give rise to all types of blood cells, including immune cells [195]. Bone marrow vasculature undergoes the same stimuli as other tissues. Thus, inflammation-induced endothelial dysfunction involves the bone marrow arteries inducing HSCs proliferation and the increased release of leukocytes into the circulation [73,186,196]. Once recruited by the activated endothelial cells, these leukocytes release cytokines and proteases, and migrate to the arterial wall, further promoting inflammation [197,198]. Epidemiological studies have shown that aging is associated with somatic (i.e., acquired) mutations in HSCs, in genes that drive the development of leukaemia [199,200]. Over time, the accumulation of these somatic mutations may lead to the progressive expansion of a mutant clone of leukocytes with altered immunological properties [201]. Approximately 10% of individuals aged 70 carry this condition. Although, they are at an increased risk of developing leukaemia; most of them never develop blood cancer [202]. Therefore, this condition is referred to as clonal haematopoiesis of indeterminate potential (CHIP) [203]. In contrast, the CHIP is associated with nearly twice the risk for CAD and ischemic stroke, independently of traditional risk factors, thus providing a new link between ageing and atherosclerosis [204,205]. Moreover, an exposure to mental stress results in the activation of a neural-hematopoietic-arterial axis, including the amygdala, bone marrow, and vascular endothelium [186,206,207]. The mental stress-induced sympathetic activation increases the bone marrow levels of noradrenaline, which promotes the HSCs proliferation, particularly myeloid cells, leading to an enhanced release of leukocytes into the circulation [208].

## 2. Inflammation Targeted Therapy

CV prevention is based on life-style changes, a reduction in risk factors, and lipid-lowering therapy. However, despite the optimal medical treatment and reduction in LDL-C levels, individuals with signs of atherosclerosis, especially in older age or with comorbidities, remain at a high risk for acute CV events [46]. Observational studies have shown that individuals with rheumatic disease, which are characterized by elevated levels of circulating cytokines, have a lower risk of atherosclerotic complications when treated with a specific anti-inflammatory therapy [62]. Moreover, the beneficial effects of statins in reducing the CV risk are due to both the reduction in cholesterol levels and inflammation inhibition [31,33,209]. More recently, the relevance of inflammatory and immune systems in the development and progression of CVD has stimulated the search for a specific systemic anti-inflammatory blocking of the cytokines pathways (Table 2) [210,211].

***Canakinumab,*** a monoclonal IL-1β antibody, was tested in patients with a high CV risk and an elevated high-sensitive C-reactive protein (hsCRP) in the Canakinumab Anti-Inflammatory Thrombosis Outcomes Study (CANTOS). The treatment significantly reduced the occurrence of a non-fatal myocardial infarction (MI), a non-fatal stroke, or a CV death [212]. The concentrations of hsCRP were also reduced, while the lipid levels did not change [41,213]. However, canakinumab was also associated with a small but statistically significant increase in the risk of fatal infections, probably due to its immunosuppressive effect.

***Methotrexate***, a broad immunosuppressive agent successfully used in patients with rheumatoid arthritis, has been tested in patients with type 2 diabetes or metabolic syndrome, in the Cardiovascular Inflammation Reduction Trial (CIRT). Low-dose methotrexate did not reduce a non-fatal MI, a non-fatal stroke, or a CV death, nor the levels of hsCRP [214].

***Colchicine*** inhibits the activation of IL-1β and the migration of leucocytes to sites of inflammation [215]. It is commonly used for the treatment of gout, pericarditis, and familial Mediterranean fever [216,217]. The Colchicine Cardiovascular Outcome Trial (COLCOT), and Low-Dose Colchicine (LoDoCo and LoDoCo2) randomized, double-blind trials, tested its anti-inflammatory effect on the CV risk in both acute and chronic CAD. Low-dose colchicine significantly reduced the risk of MI and a non-cardioembolic ischemic stroke, as well as the inflammatory markers [218,219,220,221]. However, patients treated with colchicine showed a tendency toward greater incidence gastro-intestinal adverse effects, pneumonia, and death from non-CV causes, although the difference compared to the placebo group was not statistically significant [222]. Moreover, two small-sized trials failed to show the beneficial outcomes in patients with acute coronary syndromes [223,224]. 

***Statins***. All the anti-inflammatory drugs were tested in addition to optimal therapy, including statins, which are known to exert anti-inflammatory effects in addition to lipid-lowering. Imaging studies, using ^18^F-fluorodeoxyglucose (^18^F-FDG), a glucose analogue that accumulates in the atherosclerotic lesions in proportion to the macrophage concentration and serial intravascular ultrasound, have shown that statins reduce coronary plaque inflammation, expressed by a decrease in the plaque necrotic core and volume, closely associated with a reduction in hsCRP, independently of the changes in LDL-C [225,226].

Overall, the meta-analyses of the randomized trials indicate that anti-inflammatory interventions in patients with coronary artery disease, who were already on statin therapy, reduce the risk of myocardial infarction and stroke [39,227,228]. However, their potential use is limited by the increased incidence of infections and non-CV death. Hence, there is a need to select high-risk patients who could benefit from a potentially harmful treatment. Although targeting inflammation upstream seems promising, therapies acting systemically may have important negative side effects. A further improvement will be the use of nanotechnology to produce nanoparticle drug formulations which can be delivered to specific tissues or cell populations [229].

## 3. Conclusions

Recent epidemiological, clinical, and imaging studies support the hypothesis of systemic inflammatory and immune pathogenesis of atherosclerosis and its clinical complications. The detection of features of atherosclerotic plaque vulnerability for the prediction of major CV events has limited clinical relevance. A comprehensive appraisal of atherosclerosis should shift from an anatomical imaging evaluation of atherosclerotic lesions to a qualitative assessment of a patient’s vulnerability. The current markers of inflammation have a low specificity, poorly reflecting the underlying biological processes. Hence, a combination of circulating, cellular, and imaging markers, measured over time, may identify individuals at a high risk of CVD, thus modulating specific treatments. Targeting inflammation upstream, and modulating the early phases of atherosclerosis development, may become an effective therapeutic approach. A more advanced understanding of atherosclerosis inflammatory pathways may lead to designing more specific treatments, without compromising the immune system defence against pathogens.

## Figures and Tables

**Figure 1 ijms-23-12906-f001:**
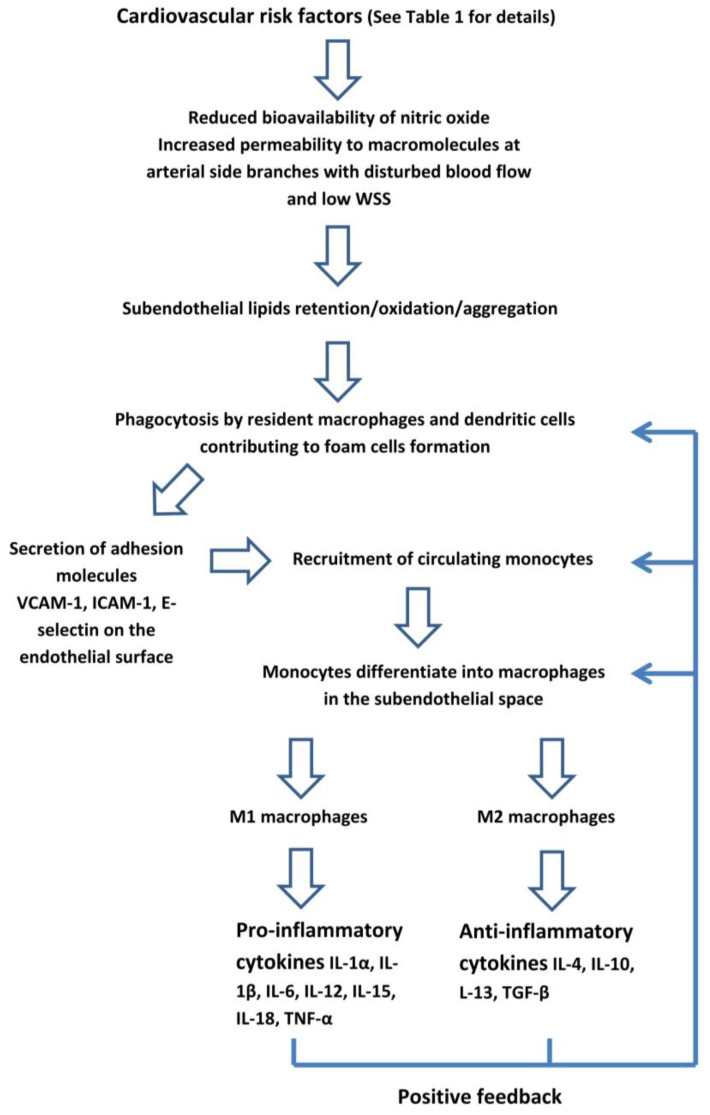
The early phase of atherosclerotic lesions starts with endothelial dysfunction which triggers a low-grade inflammatory response. WSS: wall shear stress. VCAM: vascular cell adhesion molecule. ICAM: intercellular adhesion molecule.

**Figure 2 ijms-23-12906-f002:**
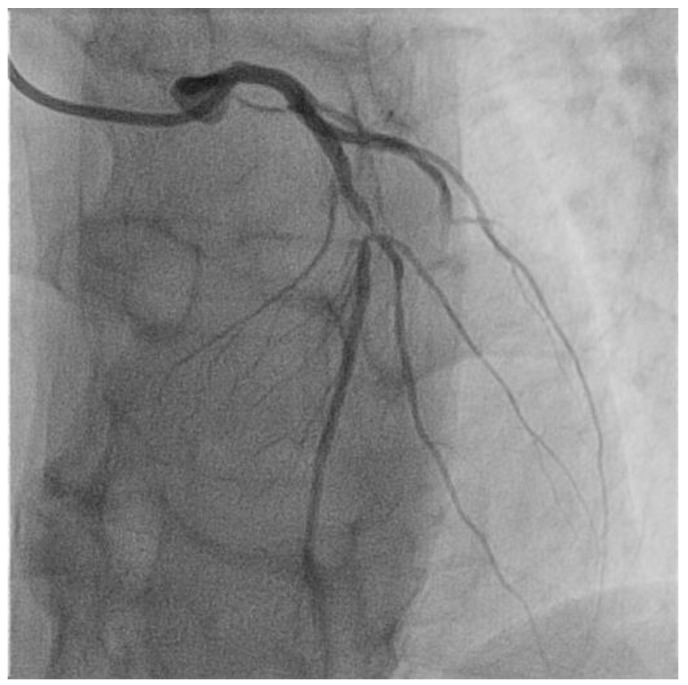
Selective angiography of the left coronary artery. Severe focal atherosclerotic narrowing of the proximal segment of left anterior descending coronary artery.

**Figure 3 ijms-23-12906-f003:**
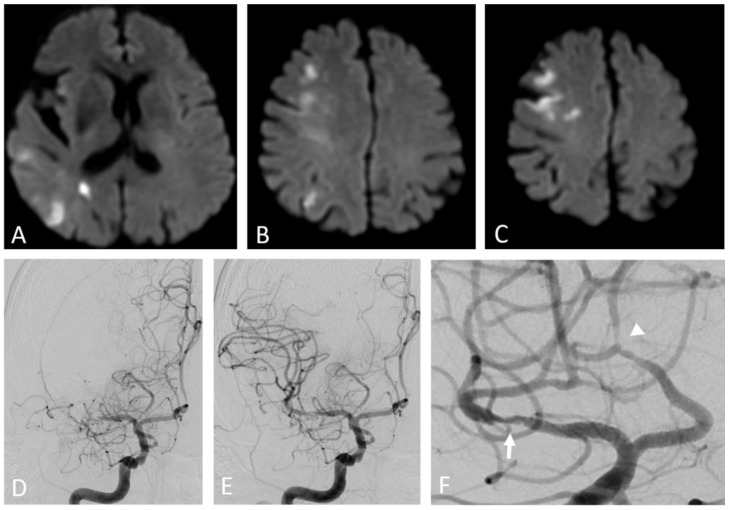
Artery-to-artery embolism and in situ thrombotic occlusion of the middle cerebral artery due to intracranial atherosclerotic disease. Hypertensive 67-year-old man. Diffusion-weighted magnetic resonance imaging demonstrates multiple cortical-subcortical ischemic lesions in the territory of the right middle cerebral artery (**A**–**C**). Digital subtraction angiography shows an occlusion of the M1 segment of the right middle cerebral artery (**D**) and its complete recanalization after mechanical thrombectomy (**E**). The magnified oblique projection after the recanalization (**F**) reveals an underlying atherosclerotic plaque at the site of the previous occlusion (arrow) and additional stenotic lesions along the course of the right anterior cerebral artery (arrowhead).

**Figure 4 ijms-23-12906-f004:**
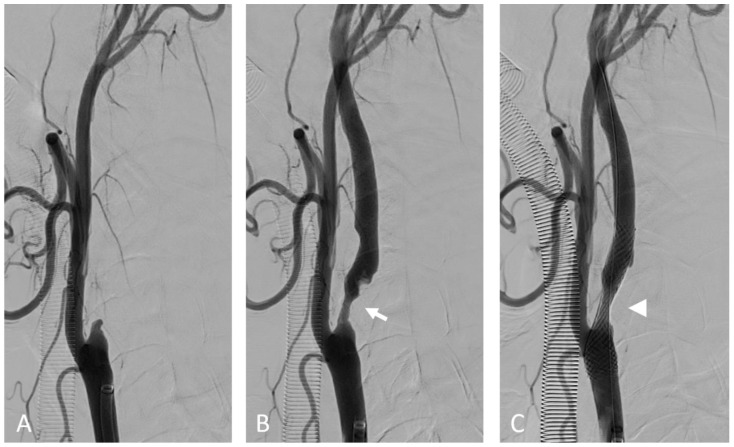
Internal carotid artery occlusion, digital subtraction angiography: acute thrombotic occlusion of the left internal carotid artery (**A**) causing sudden neurologic deficit in a 77-year-old patient. The serigraphy performed after mechanical thrombectomy and recanalization of the artery (**B**) highlights an ulcerated atherosclerotic plaque of the carotid bulb (arrow). After the administration of intravenous boluses of antiplatelet agents and heparin, a self-expanding stent (arrow) was placed in correspondence to the ulcerated plaque (**C**).

**Figure 5 ijms-23-12906-f005:**
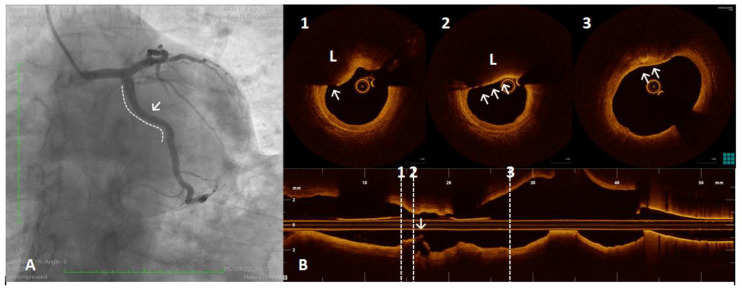
A 61 year old man referred for Non-ST-segment elevation myocardial infarction (NSTEMI). Coronary angiography showed no significant coronary lesions. Left circumflex showed haziness at the proximal segment (panel (**A**), white arrow). The OCT pull-back showed a fractured plaque (panel (**B**), 1, white arrow) associated with a lipid pool (panel (**B**), 1 and 2, “L”); thin-cap fibro-atheroma and active macrophages are easily detected because of their typical bright line (2, white arrows) or spot images, within a fibro-lipidic plaque (3, white arrows).

**Figure 6 ijms-23-12906-f006:**
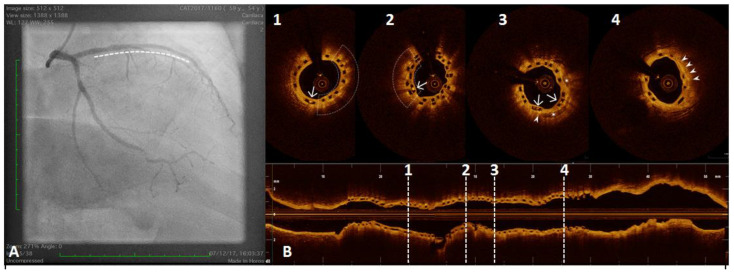
A 54 year old woman after a scheduled angiogram and OCT pull-back 3 years after an acute coronary syndrome. During index procedure 2 bioresorbable vascular scaffolds (BVS) were implanted. Coronary angiography showed satisfactory angiographical result (panel (**A**), white line). OCT pull-back confirmed a complete struts coverage and an acceptable lumen area (panel (**B**)), even at the overlapping site (2). Black boxes are typical OCT images of BVS (1,2,3 and 4). Some black box inclusions are detected (1,2 and 3 – white arrows) representing scaffold reabsorption processes. Some areas immediately below the boxes showed bright spots indicating inflammation with macrophages activation (1 and 2 - underlined area). These bright spots (3 and 4, white triangle) have been detected closely to calcium arch (3, stars) and in a fibro-lipidic plaque (4).

**Figure 7 ijms-23-12906-f007:**
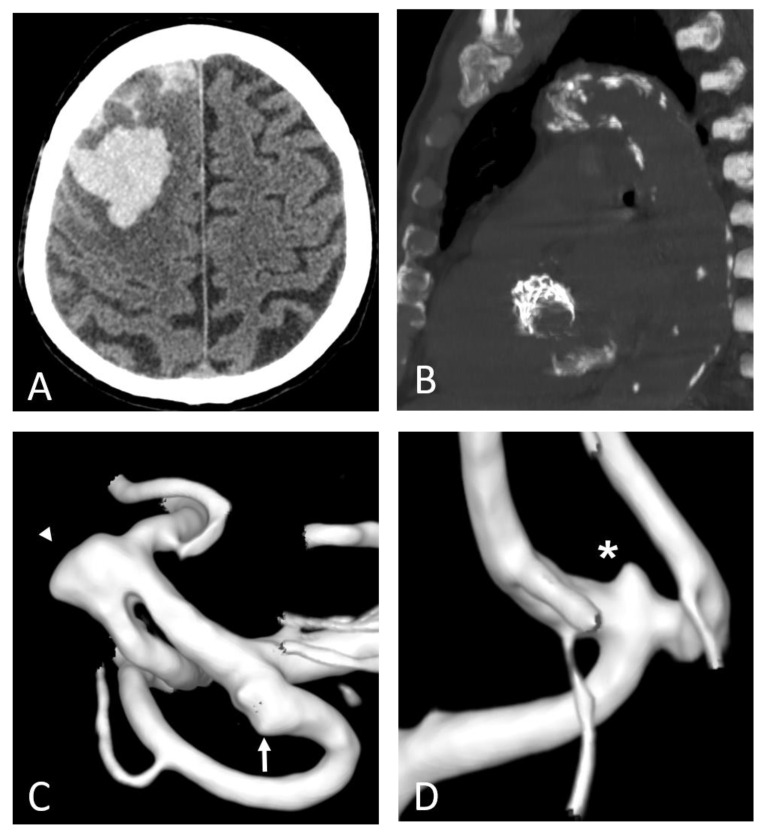
Polyvascular disease: right frontal lobar haemorrhage due to cerebral amyloid angiopathy (**A**) in an 82-year-old man with a prosthetic aortic valve and extensive calcifications of the aortic arch and along the descending thoracic aorta (**B**). Three-dimensional digital subtraction angiography reconstructions of the same patient (**C**,**D**) show diffuse arterial dysplasia with ecstatic origin of an inferior temporal branch (arrow), a dysmorphic aneurysm of the Sylvian bifurcation of the middle cerebral artery (arrowhead), and an infra-millimetric aneurysm of the anterior communicating artery (*).

**Figure 8 ijms-23-12906-f008:**
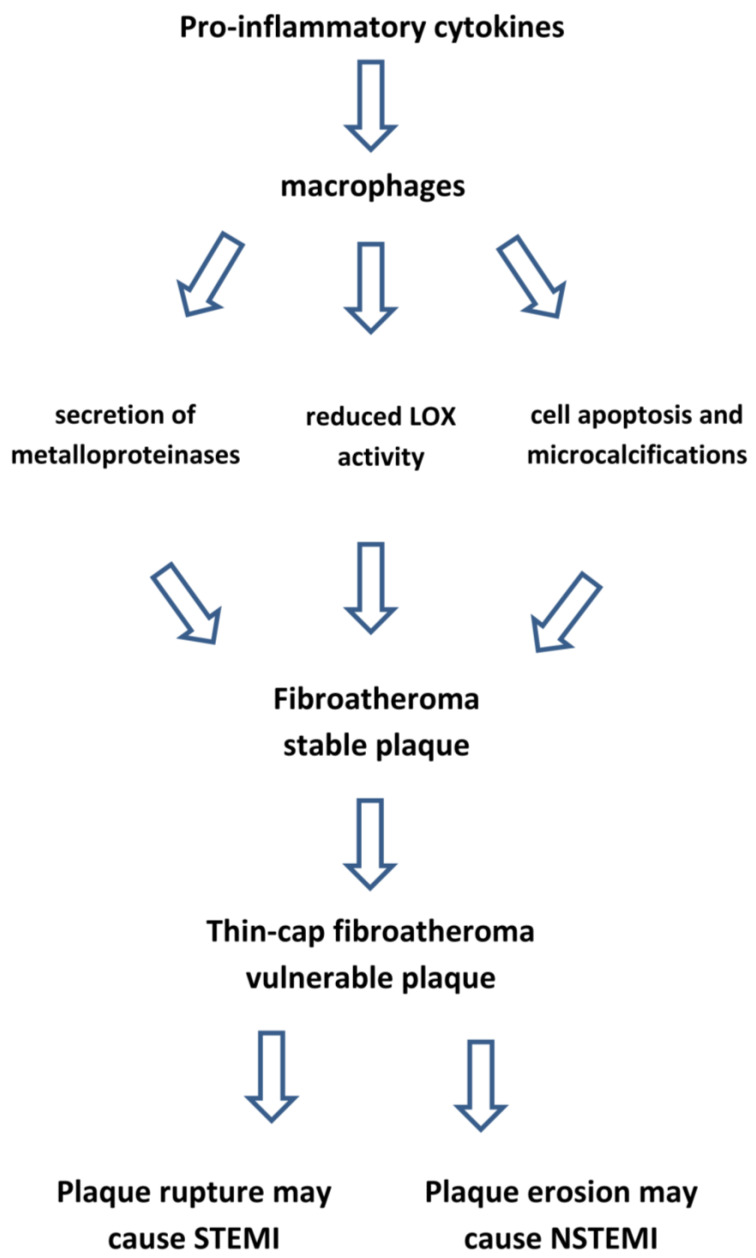
Inflammation makes atherosclerotic plaque vulnerable. LOX: lysyl oxidase enzyme. STEMI: ST-segment elevation myocardial infarction. NSTEMI: Non-ST-segment elevation myocardial infarction.

**Table 1 ijms-23-12906-t001:** Triggers of inflammatory response leading to atherosclerosis.

Traditional CV risk factors	low-density lipoprotein cholesterol triglyceride-rich lipoproteins hypertension, smoking, physical inactivity diabetes, obesity
Chronic and acute mental stress	autonomic nervous system
Ageing	bone marrow activation and clonal hematopoiesis
Chronic autoimmune diseases	rheumatoid arthritis, systemic lupus erythematosus, psoriasis, inflammatory bowel disease
Chronic infections	periodontitis, bronchitis
Acute infections	urinary tract infections, endotoxins from gut microbiota
Viral infections	influenza, COVID-19 viruses
Tissue injury	myocardial infarction, non-healing skin ulcers

See text for references.

**Table 2 ijms-23-12906-t002:** Anti-inflammatory therapies specifically blocking cytokines pathways.

Drug	Trial	Anti-Inflammatory Effect	Sample Size	Study Patients	Primary End Point	Outcome	Adverse Effects
**Canakinumab**	CANTOS [192]	interleukin-1β Inhibition	10,061	previous MI	non-fatal MI non-fatal stroke, CV death	reduced hsCRP, IL-6 −17% in primary end points	higher incidence of fatal infections
**Methotrexate**	CIRT [194]	Replication inhibition of B cells, T cells neutrophils, monocytes	4786	previous MI and T2 diabetes metabolic syndrome	non-fatal MI non-fatal stroke CV death	no change in hsCRP, IL-6, IL-1β no reduction in primary end points	increased liver enzymes reduced leukocytes
**Colchicine**	COLCOT [198]	inhibition of microtubule polymerization reduced IL-1β, IL-6	4745	1 month after MI	CV death, MI stroke	−23% in primary endpoints	diarrhoea, pneumonia
	LoDoCo2 [201]		5522			−31% in primary endpoints	increased death from non-CV causes

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
