# Peer review of "The Role of Inflammation in Cardiovascular Disease"

_ijms, 2022, doi:10.3390/ijms232112906_

Round 1

Reviewer 1 Report

In this review article the authors review the current understanding about the role of inflammation in cardiovascular disease.

Major comment:

Given the fact that to date more than 119,000 papers can be found in PubMed by searching the topics ‘inflammation’ and ‘cardiovascular disease’ it is important to state the focus of the review article more precisely and to give any information how the 208 citations used in this review were selected from the literature. Mainly, the review reads as an interest textbook article rather than a review article in a Journal dealing with molecular science. In this aspect one would have expected more details how specific pro-inflammatory cytokines are involved in the process.

Some specific points are addressed:

Mismatch between title and abstract content as large part of the abstract are not related to inflammation. Avoid abbreviation in the abstract. Condense the abstract to 200 words and give the key information rather than a summary of the whole manuscript.

Explain abbreviations when first mentioned (i.e., WSS in line 127) although it was explained in the abstract. Abstract is not part of the text-body.

Line 130: Define the endothelial receptors that sense laminar flow.

Line 134: Name the miRs that trigger this process.

Lien 135, line 152: Please name the pro-inflammtory cytokines that are released by endothelial and smooth muscle cells. What about the reduction of anti-inflammatory cytokines (such as IL-10)?

Line 160: Does the hsCRP assay monitor pro-inflammatory stage or TRL and RLP (please define!)

Line 167: Do you mean that 18F-FDG is uptaken by bone marrow and spleen and if yes, what does this mean? Why< is this uptake specific for macrophages? The next sentence on EMT has no relationship to this. Please clarify!

Line 171:  Replace ‘leukocytes’ by the term ‘monocytes’ which is more precisely as precursors of macrophages.

All the figures are just anatomical details that are interesting but with little connection to the text body. Reader expects schematic draws that visualize the key-processes and cells rather than diagnostic pictures.

Line 220: I guess that monocytes infiltrate into the tissue and differentiate into macrophages. Or do you mean that macrophages from other parts of the vascular wall migrate into the plaque area?

Line 257: Please consider the role of lysyl oxidase that adds to the modification of the extracellular matrix. MMPs do not the whole job. Name the MMPs more specifically. Some are specific for macrophages (MMP12). Does this matter?

Chapter ‘vulnerable plaques’ (line 265-284): Although this is an interesting point, the question in the context of this review is whether inflammation plays a role in the variability of plaque architecture in one artery and how this may be explained. Please link this chapter to the topic of the review.

Conclusion from clinical studies: Is there any predicted candidate that will have an impact in the future?

Clarify the molecular concepts by which statins act anti-inflammatory without the lipid-lowering effect. What are the targets? As this is the most promising treatment it is necessary to make this point clear.

Author Response

The role of inflammation in cardiovascular disease

Responses to Reviewers, 15 October 2022

We are thankful for the reviewers’ efforts in reviewing our manuscript and for the valuable suggestions which have strengthened it. We hereby address the raised response to the comments.

Reviewer 1

Reviewer comment. Mismatch between title and abstract content as large part of the abstract are not related to inflammation. Avoid abbreviation in the abstract. Condense the abstract to 200 words and give the key information rather than a summary of the whole manuscript

Authors response. The Abstract has been condensed in 235 words, strictly related to inflammation, without abbreviations.

Reviewer comment. Explain abbreviations when first mentioned (i.e., WSS in line 127) although it was explained in the abstract. Abstract is not part of the text-body.

Authors response. Abbreviation included when first mentioned

Reviewer comment. Line 130: Define the endothelial receptors that sense laminar flow.

Authors response. A sentence has been added: Specific endothelial biomechanical receptors such as glycocalyx, a proteoglycan layer which covers the apical surface of the endothelial cells, sense and distinguish laminar and non-uniform patterns of blood flow, translating WSS into biochemical signals.

Reviewer comment. Line 134: Name the miRs that trigger this process.

Authors response. Names added: Conversely, decreased WSS induces the expression of endothelial genes, controlled by flow-responsive endothelial microRNAs (miRNA), such as miRNA 92a, 663, 712,

Reviewer comment. Line 135, line 152: Please name the pro-inflammatory cytokines that are released by endothelial and smooth muscle cells. What about the reduction of anti-inflammatory cytokines (such as IL-10)?

Author response. Names included: Once entering the subendothelial space, the recruited leukocytes, especially monocytes, differentiate into macrophages and then polarize, adopting different functional phenotypes, in response to their microenvironment.70 T lymphocytes activate these cells into pro-inflammatory M1 macrophages, which elaborate pro-inflammatory cytokines (interleukin IL-1α, IL-1β, IL-6, IL-12, IL-15, IL-18, and tumour necrosis factor (TNF)-α) involved in atherosclerosis progression, or alternative anti-inflammatory M2 macrophages which elaborate anti-inflammatory cytokines (IL-4, IL-10, IL-13, and transforming growth factor (TGF)-β) which have a critical role in the resolution of inflammation and plaque healing.71-75 Some interleukins (IL-1β, IL-6, and IL-12) control the hepatic production of C-reactive protein (CRP), an important biomarker of CV risk.76-79 Although macrophages are the main source of cytokines, other cells such as lymphocytes, endothelial cells, and polymorphonuclear leukocytes contribute to their production.

Reviewer comment. Line 160: Does the hsCRP assay monitor pro-inflammatory stage or TRL and RLP (please define!)

Authors response. The sentence has been made clearer: Moreover, in recent years, the large increase in the prevalence of type 2 diabetes and obesity, and the control of LDL-C with effective treatment, shifted the lipid risk profile in the population from elevated LDL-C to elevated triglyceride-rich lipoproteins (TRL) and remnant lipoproteins (RLP), more strongly associated with inflammation than LDL-C.

Reviewer comment. Line 167: Do you mean that 18F-FDG is uptaken by bone marrow and spleen and if yes, what does this mean? Why< is this uptake specific for macrophages? The next sentence on EMT has no relationship to this. Please clarify!

Authors response. The paragraph has been updated: The endothelial inflammatory response includes the coordinate activation of both innate immunity (macrophages) and adaptive immunity, (T- and B-lymphocytes, dendritic cells).46,68 Leuocytes involvement in inflammation and atherosclerosis has also been shown by human positron emission tomography (PET) studies, using 18F-fluorodeoxyglucose (18F-FDG) a glucose analogue, extensively used as a marker of metabolic activity for malignancy staging. It is used in vascular inflammation imaging because it accumulates mostly in macrophages due to their high glucose metabolic activity, especially after inflammatory activation.69,70 Increased uptake has also been found in bone marrow and spleen of patients with CAD compared with those without. This confirms the association between bone marrow and spleen hematopoietic activation and increase in proinflammatory mediators involved in atherosclerotic plaque inflammation.71-73

Reviewer comment. Line 171:  Replace ‘leukocytes’ by the term ‘monocytes’ which is more precisely as precursors of macrophages.

Authors response. The sentence has been changed: Once entering the subendothelial space, the recruited monocytes differentiate into macrophages and then polarize, adopting different functional phenotypes, in response to their microenvironment

Reviewer comment. All the figures are just anatomical details that are interesting but with little connection to the text body. Reader expects schematic draws that visualize the key-processes and cells rather than diagnostic pictures.

Authors response. Schematic draws showing the role of inflammation in initiating atherosclerosis and progression to vulnerable plaque, have been added

Reviewer comment. Line 220: I guess that monocytes infiltrate into the tissue and differentiate into macrophages. Or do you mean that macrophages from other parts of the vascular wall migrate into the plaque area?

Authors response. This statement has been clarified: If inflammation persists, there will be subsequent cycles of monocytes infiltration which differentiate into macrophages, that undergo death, leading to microcalcification development.106,107 Along with TCFA and macrophage, microcalcifications strongly contribute to plaque instability, especially when they co-localize with macrophages in the same plaque (reciprocal distance less than 100 µm), as demonstrated by optical coherence tomography (OCT).108-111

Reviewer comment. Line 257: Please consider the role of lysyl oxidase that adds to the modification of the extracellular matrix. MMPs do not the whole job. Name the MMPs more specifically. Some are specific for macrophages (MMP12). Does this matter?

Authors response. Thank you for your suggestion. A paragraph has been added: Also, the stability of the fibrous cap depends on collagen fibre cross-linking, which is modulated by the enzyme lysyl oxidase (LOX) expressed by the endothelial cells.137,138 High LOX levels are associated with plaque stability and the healing process within the plaque.139 Endothelial dysfunction induced by CV risk factors and mediators of inflammation, such as macrophages derived cytokines, reduce LOX activity resulting in abnormal collagen cross-linking. This process weakens the fibrous cap and increases the soluble forms of collagen which may undergo MMP degradation.

Reviewer comment. Chapter ‘vulnerable plaques’ (line 265-284): Although this is an interesting point, the question in the context of this review is whether inflammation plays a role in the variability of plaque architecture in one artery and how this may be explained. Please link this chapter to the topic of the review.

Authors response. This is an interesting point. The paragraph “Vulnerable plaque” has been merged with “Extracellular matrix”, with new additions showing the role of inflammation in the plaque vulnerability: Vulnerable plaque. Inflammation is a critical feature of vulnerable plaques, although lesser degrees of inflammation have also been found in stable ones.130 Atherosclerotic plaques consist largely of extracellular matrix (ECM), including collagen, elastin, proteoglycan, and glycosaminoglycan, synthesized by smooth muscle cells in the arterial wall. ECM is interlinked with plaque calcification, both contributing to plaque stability.131 Microcalcification localize between collagen fibers and regions lacking collagen, so that the proportions of ECM and microcalcification are inversely related. Although interstitial collagen is a plaque stabilizing factor, it also contributes to accumulation of lipoprotein particles within the arterial wall.115,132 In conditions of inflammation, cytokines (IL-1β, TNF-α) induce the secretion of metalloproteinases (MMPs), especially MMP-1, MMP-8, MMP-9, MMP-12, MMP-13, from macrophages, controlled by microRNAs.133-135 MMPs catalyse the breakdown of the interstitial collagen resulting in thinning and weakening of the fibrous cap, thus compromising its tensile strength and making the plaque unstable.136

Also, the stability of the fibrous cap depends on collagen fibre cross-linking, which is modulated by the enzyme lysyl oxidase (LOX) expressed by the endothelial cells.137,138 High LOX levels are associated with plaque stability and the healing process within the plaque.139 Endothelial dysfunction induced by CV risk factors and mediators of inflammation, such as macrophages derived cytokines, reduce LOX activity resulting in abnormal collagen cross-linking. This process weakens the fibrous cap and increases the soluble forms of collagen which may undergo MMP degradation. Statins have been shown to inhibit the secretion of MMPs from inflammatory cells and normalize endothelial LOX expression, thus increasing plaque collagen.140-144 Therefore, in addition to the increase in the calcium content of the atherosclerotic plaque, these effects account for the plaque stabilization induced by statins.

Reviewer comment. Conclusion from clinical studies: Is there any predicted candidate that will have an impact in the future?

Authors response. Sentence added: Although targeting inflammation upstream seems promising, therapies acting systemically may have important negative side effects. Further improvement will be the use of nanotechnology to produce nanoparticles drug formulations which can be delivered to specific tissues or cell populations.228

Reviewer comment. Clarify the molecular concepts by which statins act anti-inflammatory without the lipid-lowering effect. What are the targets? As this is the most promising treatment it is necessary to make this point clear

Authors response. Statins targets accounting for plaque stabilization: Statins have been shown to inhibit the secretion of MMPs from inflammatory cells and normalize endothelial LOX expression, thus increasing plaque collagen.140-144 Therefore, in addition to the lipid-lowering effect and increase in the calcium content of the atherosclerotic plaque, these anti-inflammatory effects account for  plaque stabilization induced by statins.

Reviewer 2 Report

In this paper, the authors analyzed the involvement of the inflammation in the onset and progression of atherosclerosis. Nevertheless, the aim of the review is not clear and the contributi9n to the field is very poor. Unfortunately, I think that the review, in this form, is not eligible  for pubblication. Below some suggestions to improve the paper.

Abstract has to be shorter and gives the main information about the topic of the paper

I would give the essential information about the key actors of atherosclerosis pathophysiology and spend further words on the pharmacological applications

I would add figures or tables useful to summarize main concepts of the paper

Author Response

The role of inflammation in cardiovascular disease

Responses to Reviewers, 15 October 2022

We are thankful for the reviewers’ efforts in reviewing our manuscript and for the valuable suggestions which have strengthened it. We hereby address the raised response to the comments.

Reviewer #2

Reviewer comment. Abstract has to be shorter and gives the main information about the topic of the paper

Authors response. Abstract has been shortened and re-organized to give the main information about the topic of the paper

Reviewer comment. I would give the essential information about the key factors of atherosclerosis pathophysiology and spend further words on the pharmacological applications

Authors response. Several paragraphs have been expanded to give details about pathophysiology of inflammation in atherosclerosis development. The rationale for discussing inflammation targeted therapy has been added to pharmacological treatment

Reviewer comment. I would add figures or tables useful to summarize main concepts of the paper

Authors response. Schematic draws summarizing the inflammatory factors in atherosclerosis development and progression to vulnerable plaque have been added

Reviewer 3 Report

Questions: In general, thanks to the author for an interesting review.   However, there were a few remarks:   1. In the review, the authors present several drawings and photographs. Who owns these images, the authors?   2. We need a general resulting scheme of atherosclerosis risk factors.   3. It is not entirely clear what is the essence of the review?   4. What is the personal opinion of the authors about the disclosed problem? what problems does it pose for further research?   4. There is no novelty in some points of observation, for example: inflammation in inflammation of coronary plaques, immune response to inflammation.....   5. I recommend expanding the conclusion, as it does not reflect all the points presented in the review.

Author Response

The role of inflammation in cardiovascular disease

Responses to Reviewers, 15 October 2022

We are thankful for the reviewers’ efforts in reviewing our manuscript and for the valuable suggestions which have strengthened it. We hereby address the raised response to the comments.

Reviewer #3

Reviewer comment. 1. There are several figures in the review. It is not entirely clear whose drawings, the authors? It would be possible to recommend to the authors to present schematically the participation of each factor in the pathogenesis of atherosclerosis.

Authors response. All figures derive from the personal clinical practice of the authors. Schematic draws summarizing the inflammatory factors in atherosclerosis development and progression to vulnerable plaque have been added

Reviewer comment. We need a general resulting scheme of factors.

Authors response. This has been added to the text

Reviewer comment. It is not entirely clear what is the essence of the review? What is the personal opinion of the authors on the problem under study? what problems does it pose for further consideration?

Authors response. The present paper is a Review not a research paper, it summarizes the state of the art of the our understanding of the mechanisms and role of inflammation in plaque pathophysiology. The authors’ personal opinion is indirectly expressed through the data presented and discussion.

Reviewer comment. 4. No novelty in some points of review, for example: inflammation in the development of coronary plaques, immune response to inflammation.....

Authors response. Several paragraphs have been modified and expanded, discussing in more details the mechanisms of inflammation leading to the development of atherosclerotic plaques.

Reviewer comment. I would recommend expanding the output. In its current form, it does not reflect the current state of affairs.

Authors response. We acknowledge your recommendation. Several paragraphs have been modified and expanded.

Reviewer 4 Report

Henein et al. reviewed the role of systemic and local inflammation in the development and progression of atherosclerosis, cardiovascular disease, and coronary and cerebral artery disease. The authors also discussed several clinical trials of anti-inflammatory therapy.

1.     The authors should cut down some words of the abstract, make it more precise, and be sure to follow the word limit requirement by IJMS.

2.     The manuscript needs a re-organization or new section titles and transition sentences to help tell the story. For now, it is not very clear in what order the context in the introduction section follows. The authors should also describe their brief plan in one or two sentences about how the article is organized at the end of the introduction paragraph.

3.     The paragraph title Immune response to inflammation (line 139) sounds weird, or do you mean ‘immune response during inflammation? The title Extracellular Matrix (line 250) is also not descriptive, the authors should consider renaming these paragraphs.  

4.     Authors should expand the discussion on aging-activated inflammation (line 346) and consider adding a paragraph for the viral infection-related inflammatory response to CVD.

5.     The rationale for discussing inflammation-targeted therapy in this review should be expanded in section 2. This content should be summarized in the conclusion section.

6.     In table 2, the drug title is missing on the last row. Authors may either name it as low-dose colchicine or reformat the table. Trial information should also be filled up.

7.     It is generally weird to use bold text in the middle of a sentence. If authors want to emphasize the bold text is a start of a new section, a separate title should be added at the beginning of that paragraph. Bold text (lines 378, 385, 388, 400).

8.     The authors should fix the following formatting issues and typos: extra space (lines 157 and 253) and typo (line 251).

Author Response

The role of inflammation in cardiovascular disease

Responses to Reviewers, 15 October 2022

We are thankful for the reviewers’ efforts in reviewing our manuscript and for the valuable suggestions which have strengthened it. We hereby address the raised response to the comments.

Reviewer #4

Reviewer comment. 1. The authors should cut down some words of the abstract, make it more precise, and be sure to follow the word limit requirement by IJMS.

Authors response. The Abstract has been condensed in 235 words

Reviewer comment. 2.The manuscript needs a re-organization or new section titles and transition sentences to help tell the story. For now, it is not very clear in what order the context in the introduction section follows. The authors should also describe their brief plan in one or two sentences about how the article is organized at the end of the introduction paragraph

Authors response. The manuscript has been re-organized: the paragraph “Vulnerable plaque” has been merged with “Extracellular matrix”, with new additions showing the role of inflammation in the plaque vulnerability. The paragraph “Immune response to inflammation” merged with “Endothelial dysfunction”.

The “Introduction” has been largely modified, also briefly describing the plan of the Review:

Over the last two decades, clinical and experimental studies have shown that atherosclerosis is a low-grade, sterile, inflammatory disease.1,2 Systemic and local inflammation have a central role in the development and progression of cardiovascular disease (CVD), from endothelial dysfunction to clinical syndromes.3-6 Inflammatory biomarkers have been shown to predict CVD, independently of traditional risk factors.7-9 Several acute and chronic conditions, including traditional risk factors, psychological stress, autoimmune disease, microbial and viral infections, and ageing, can activate endothelial damage and dysfunction (Table 1).10-24 In turn, this promotes vascular low-grade inflammatory response leading to progression of atherosclerosis.25 Hence, inflammation is a common mechanism linking traditional and emerging CV risk factors to the development of atherosclerosis, leading to CAD, large artery thrombotic stroke, and cerebral aneurysms.1,26-29 All phases of atherosclerosis, from retention of atherogenic lipoproteins within the arterial wall, to plaque development and rupture, involve a complex network including innate and adaptive immune systems, bone marrow and spleen, which modulate pro-inflammatory and anti-inflammatory activities of protein mediators, such as cytokines, and immune cells such as leukocytes, macrophages, and lymphocytes. The role of inflammation in the atherosclerosis is confirmed by the effects of statins in reducing the CV risk. Several studies have shown that most beneficial effects of statins are due to reduction of vascular inflammation, to some extent independent of their lipid-lowering action.30-32 Moreover, nearly one-half of patients undergoing high-intensity lipid-lowering treatment with statins in secondary prevention trial, have residual inflammatory risk and increased risk of major CV events, despite significant lipid-lowering effects.33-37 In the last few years, the inflammatory biology of atherosclerosis has been translated into therapeutic strategies. Recent clinical trials indicated that targeting inflammation results in lower incidence of CAD and stroke.38-40 This review summarizes current knowledge about the role of inflammation and the immune systems in the development of atherosclerosis, progression to stable and vulnerable plaque, the relationship between central nervous system and arterial inflammatory response, the role of ageing in promoting atherosclerosis beyond prolonged exposure to traditional risk factors, and new therapeutic opportunities targeting inflammation to reduce CVD burden. Although most studies refer to CAD, the relationship between inflammation and atherosclerosis in coronary and cerebral arteries is based on the same mechanisms.1,27,41

Reviewer comment. 3. The paragraph title Immune response to inflammation (line 139) sounds weird, or do you mean ‘immune response during inflammation? The title Extracellular Matrix (line 250) is also not descriptive, the authors should consider renaming these paragraphs.

Authors response. Paragraphs have been re-organized (please, see point 2): the paragraph “Vulnerable plaque” has been merged with “Extracellular matrix”, with new additions showing the role of inflammation in the plaque vulnerability. The paragraph “Immune response to inflammation” merged with “Endothelial dysfunction”.

Reviewer comment. 4. Authors should expand the discussion on aging-activated inflammation (line 346) and consider adding a paragraph for the viral infection-related inflammatory response to CVD.

Authors response. The paragraph on aging-activated inflammation focuses on clonal hematopoiesis, while the discussion of the interesting aging-related inflammatory changes would have deserved a whole review. A specific paragraph on inflammatory response to viral infection is lacking because mechanisms are very similar to those induced by risk factors and other infections.

Reviewer comment. 5.The rationale for discussing inflammation-targeted therapy in this review should be expanded in section 2. This content should be summarized in the conclusion section.

Authors response. The rationale for discussing inflammation targeted therapy has been added: Inflammation targeted therapy

CV prevention is based on life-style changes, reduction of risk factors, and lipid-lowering therapy. However, despite optimal medical treatment and reduction of LDL-C levels, individuals with signs of atherosclerosis, especially in older age or with comorbidities, remain at high risk for acute CV events.45 Observational studies have shown that individuals with rheumatic disease, which are characterized by elevated levels of circulating cytokines, have lower risk of atherosclerotic complications when treated with specific anti-inflammatory therapy.61 Moreover, the beneficial effects of statins in reducing CV risk, are due to both reduction of cholesterol levels and inflammation inhibition.30,32,209 More recently, the relevance of inflammatory and immune systems in the development and progression of CVD has stimulated the search for specific systemic anti-inflammatory blocking of the cytokines pathways (Table 2).210,211

The content of Inflammation targeted therapy has been summarized in the Conclusions: Targeting inflammation upstream, modulating the early phases of atherosclerosis development, may become an effective therapeutic approach. A more advanced understanding of atherosclerosis inflammatory pathways may lead to designing more specific treatments, without compromising the immune system defence against pathogens.

Reviewer comment. 6. In table 2, the drug title is missing on the last row. Authors may either name it as low-dose colchicine or reformat the table. Trial information should also be filled up.

Authors response. Table 2 has been modified adding the name of the drug and trial information

Reviewer comment. 7. It is generally weird to use bold text in the middle of a sentence. If authors want to emphasize the bold text is a start of a new section, a separate title should be added at the beginning of that paragraph. Bold text (lines 378, 385, 388, 400).

Authors response. Text has been modified according to the reviewer’s suggestion

Reviewer comment. 8. The authors should fix the following formatting issues and typos: extra space (lines 157 and 253) and typo (line 251).

Authors response. The text has been largely modified and re-organized.

Round 2

Reviewer 1 Report

I thank the authors for the improvements of the manuscript.

Reviewer 2 Report

I think that the authors have addressed the majority of my suggestions and that now the paper is strongly improved.

Reviewer 3 Report

I consider the authors' response to all comments to be more than exhaustive. Recommend for publication

Reviewer 4 Report

The authors have addressed most of my concerns in this manuscript. However, tables 1 and 2 were missing in this updated manuscript. Plus, the authors should work on the formatting as there are inconsistencies in paragraph breaks and text font.